# An Increased Risk of School-Aged Children with Viral Infection among Diarrhea Clusters in Taiwan during 2011–2019

**DOI:** 10.3390/children8090807

**Published:** 2021-09-15

**Authors:** Fu-Huang Lin, Yu-Ching Chou, Bao-Chung Chen, Jui-Cheng Lu, Chung-Jung Liu, Chi-Jeng Hsieh, Chia-Peng Yu

**Affiliations:** 1School of Public Health, National Defense Medical Center, Taipei City 11490, Taiwan; noldling@ms10.hinet.net (F.-H.L.); trishow@mail.ndmctsgh.edu.tw (Y.-C.C.); 2Division of Gastroenterology, Department of Internal Medicine, Tri-Service General Hospital, National Defense Medical Center, Taipei City 11490, Taiwan; staineely@yahoo.com.tw; 3Department of Medical Records, Tri-Service General Hospital, National Defense Medical Center, Taipei City 11490, Taiwan; zx21425798@gmail.com (J.-C.L.); na660426@gmail.com (C.-J.L.); 4Department of Health Care Administration, Asia Eastern University of Science and Technology, New Taipei City 22061, Taiwan; fl004@mail.aeust.edu.tw

**Keywords:** diarrhea, norovirus, *Staphylococcus aureus*, school, cluster, retrospective study

## Abstract

Acute diarrhea is mainly caused by norovirus and rotavirus. Numerous factors modify the risk of diarrhea cluster infections and outbreaks. The purpose of this study was to explore the epidemiological characteristics, differences, and trends in the distribution of viral and bacterial pathogens that cause diarrhea cluster events as well as the public places where diarrhea cluster events took place in Taiwan from 2011 to 2019. We examined publicly available, annual summary data on 2865 diarrhea clusters confirmed by the Taiwan Centers for Disease Control (CDC) from 2011 to 2019. There were statistically significant differences (*p* < 0.001) in event numbers of diarrhea clusters among viral and bacterial pathogens, and statistically significant differences (*p* < 0.001) in event numbers of diarrhea clusters among bacterial pathogens. There were also statistically significant differences (*p* < 0.001) in the event numbers of diarrhea clusters among public places. Norovirus infections were the first most numerous (77.1%, 1810/2347) diarrhea clusters among viral and bacterial infections. Among bacterial infections, *Staphylococcus aureus* infections accounted for the greatest number of diarrhea clusters (35.5%, 104/293). Schools were the places with the greatest number of diarrhea clusters (49.1%, 1406/2865) among various institutions. Norovirus single infection (odds ratio, OR = 4.423), *Staphylococcus aureus* single infection (OR = 2.238), and school (OR = 1.983) were identified as risk factors. This is the first report of confirmed events of diarrhea clusters taken from surveillance data compiled by Taiwan’s CDC (2011–2019). This study highlights the importance of long-term and geographically extended studies, particularly for highly fluctuating pathogens, to understand the implications of the transmission of diarrhea clusters in Taiwan’s populations. Importantly, big data have been identified that can inform future surveillance and research efforts in Taiwan.

## 1. Introduction

Acute infectious diarrhea is a common disease around the world. Infectious norovirus and rotavirus are the main cause of the disease. In industrialized countries, acute diarrhea is usually self-limiting, but the morbidity is high among children and elderly patients. In non-developed countries, viral diarrhea is a notable cause of death, especially in infants [1,2]. Viral diarrhea leads to the deaths of approximately 200,000 children around the world every year. Sporadic cases may occur, but viral diarrhea more commonly occurs in cluster infections and outbreaks within close communities such as cruise ships, nursing facilities, and daycare centers, etc.

Norovirus and rotavirus are the most common viral causes of diarrhea. Norovirus is the main pathogen that causes viral diarrhea and acute gastritis symptoms. Especially in developing countries, rotavirus is likely to cause severe acute gastroenteritis in children, and most deaths are likely to occur in these countries. Although the mortality rate caused by norovirus or rotavirus in developed countries is not high, there are still sporadic mass incidents that tend to occur in densely populated places, such as the military, cruise ships, campuses, hospitals, and nursing homes, etc. [3].

Norovirus belongs to the Caliciviridae viral family and is a positive, single-stranded, RNA virus. According to genotyping, it can be divided into at least seven genogroups (GI-GX) and can be subdivided into more than 40 genotypes [4]. The main genogroups that infect humans are GI, GII, and GIV [5]. According to statistics from US CDC, the main genogroup that infects people is the GII. 4 genotype, which has circulated since the mid-1990s [6]. In addition to the highest genetic variation, this gene often causes pandemics every 2–3 years. The threat of norovirus, to humans, is due to the following factors: (1) Norovirus genes are highly variable, and they very easily mutate and evolve new virus strains [7]; (2) The virus has a low infectious dose and a few norovirus virus particles (about 18–1000 virus particles) can infect healthy adults through human-to-human transmission [8]; (3) In patients who have been infected with norovirus, virus particles can continue to be discharged from the body through the feces (viral shedding) after the symptoms are relieved, over three weeks, which increases the risk of virus transmission [9]; (4) Norovirus particles are quite stable in the environment, and the virus particles will not be damaged even when frozen or heated to 60 °C. Therefore, norovirus infection may occur if food is raw [10]. Norovirus is known to be transmitted mainly through imbibing contaminated food or water sources, as well as through direct human-to-human contact. However, the literature also indicates that there are also other possible modes of transmission [11]. Norovirus can infect any age group, and it is more common in the adult population. The main route of infection is through fecal-oral infection. After norovirus infection, the incubation period is about 12–48 h, and symptoms can last for 12–60 h. A few patients can have symptoms for more than 3 days, but most patients can heal themselves. If the infection occurs in a child under 5 years old or an elderly person over 65 who cannot take care of themselves, serious dehydration may occur, and they may even require hospitalization.

Rotavirus belongs to the Reoviridae virus family. Its genome structure is 11 double-stranded RNA without a protein coat. Nine serotypes can be classified according to the viral protein VP6 [12,13]. Groups A to C infect humans whereas the other groups are found in animals. Rotavirus A is the most common species causing more than 90% infections in humans. The 50 P genotypes and 35 G genotypes can be identified by using molecular biological methods with the proteins VP4 and VP7 [14,15,16]. Rotavirus is an important pathogen that causes acute gastroenteritis in children under five years old [17]. The report on the global burden of the disease and several extended analyses on rotaviruses, along with the results of rotavirus vaccination, found that rotavirus infection caused 128,500 deaths and 258,173,300 episodes of diarrhea among children younger than 5 years in 2016 [18]. A recent meta-analysis by Burnett et al. indicated that rotavirus vaccines were effective in preventing rotavirus diarrhea, with higher performance in countries with lower child mortality [19]. The main symptoms of infection are vomiting, watery diarrhea, and fever. The main routes of infection are fecal-oral infection and person-to-person contact. At present, vaccination is considered the most effective preventive measure.

Diarrhea clusters are a public health issue, information is highly valued by global health authorities. Some governments have established monitoring systems and published research reports on diarrheal cluster events. For instance, The Health Protection Agency is a national surveillance scheme to monitor general outbreaks of infectious intestinal diseases [20]. Norovirus is the primary pathogen involved in diarrhea clustering in Taiwan. In February 1993, the first case of nosocomial infection caused by norovirus was identified in Taiwan [21,22]. Since then, diarrhea clustering caused by noroviruses has been a concern. The Centers for Disease Control (CDC) of Taiwan’s Ministry of Health and Welfare has begun to publish a summary of diarrhea cluster epidemics. From these data, the collection and reporting of diarrhea clusters has significantly improved in terms of type, quantity, completeness, timeliness, and comprehensiveness. The purpose of this study was to examine the past nine years of empirical data provided by the Taiwan National Infectious Disease Statistics System (TNIDSS) [23] to obtain a retrospective historical perspective of diarrhea clusters, and explore the causes, trends, and epidemiological changes that cause cluster infections.

## 2. Materials and Methods

This study did not require ethical approval as it involves information freely available in the public domain, and the analysis of open dataset sources, where the data are properly anonymized [24].

Definition of confirmed cases: (1) intestinal symptoms: diarrhea three times or more a day, and accompanied by vomiting or fever, or watery diarrhea. (2) Case definition: Excluding patients with diarrhea associated with notifiable diseases; including patients with intestinal symptoms and criteria for person, time and place that are suspected as cluster infections with concern regarding spread. The diarrhea clustering event is based on a number of persons with diarrhea at a similar time/place and patients with the same pathogenic infection. The data were organized by cluster, which contained multiple diarrhea cases, all tied to a single event. Based on the policy of Taiwan’s CDC, patients of suspected food poisoning events that report to the product management distribution system and obtained the event ID are able to submit specimens through reporting to the symptom surveillance system.

This study used the “TNIDSS public database” from the website platform of Taiwan’s CDC [23]. The TNIDSS public database includes all notifiable diseases in categories 1 to 5 as stipulated by Communicable Disease Control Act. In order to ensure information security and prevent a breach of privacy regarding any cases, the system’s database does not store any privacy-related information, only secondary data with statistical values. Taiwan’s CDC’s database includes a “Symptom Surveillance System” (diarrhea clusters). The database does not contain a medical history of patients, signs and symptoms.

The research structure of this study was based on the following method: retrospective historical study of all of the local diarrhea clusters from 2011 to 2019. We determined the number of diarrhea clustering events from 2011 to 2019, and examined the differences and trends in the distribution of their epidemiological characteristics (types of diarrhea causing pathogens and diarrhea clustering sites). We used the numbers (N) and percentages (%) for categorical variables to present pathogen detections and pathogen infections (including viruses, bacteria and other pathogens) in each year for diarrhea clusters. Chi-square tests or Fisher’s exact tests (when any expected numbers in cells were less than 5) were used to examine the differences in the distribution of the categorical variables. To estimate the effect of different types of pathogen (e.g., norovirus vs. non-norovirus, *Staphylococcus aureus* vs. non-*Staphylococcus aureus*) and different institutions (e.g., school vs. non- school) on the risk during different calendar years (e.g., in 2011 vs. other years (combinate numbers of other years)), the logistical model was the estimated odds ratio and 95% CIs. A two-tailed *p*-value < 0.05 was considered statistically significant. All statistical analyses were performed using SPSS software (IBM SPSS Statistics 21; Asia Analytics Taiwan, Taipei, Taiwan).

## 3. Results

This study investigated the statistics regarding the communicable disease and surveillance report provided by Taiwan’s CDC 2011-2019 on the number of diarrhea clusters (clusters were of 114, 145, 106, 106, 357, 452, 575, 424, and 586 events, and the total was 2865) (Table 1). The routine examination of diarrhea cluster infection detected 2196 pathogen-positive cases, 650 pathogen-negative cases, and 19 samples that were not tested.

There was a statistically significant difference (*p* < 0.001) in the event numbers of diarrhea clusters among viral and bacterial pathogens between 2011 and 2019 (Table 2).

The number of norovirus-only infection clusters was 1659, the number of rotavirus-only infection clusters was 120, the number of norovirus and rotavirus combined infection clusters was 151, the number of adenovirus infection clusters was 1, and the number of bacterial infection clusters was 265. Cases in 2012 had 4.423 times the odds of being norovirus as cases in other years. There were statistically significant differences (*p* < 0.001) in event numbers of diarrhea clusters among bacterial pathogens between 2011 and 2019 (Table 3).

The number of diarrhea clusters caused by pathogenic bacteria from 2011 to 2019 was 8, 3, 1, 2, 39, 30, 44, 62, and 76, respectively (265 in total). Cases in 2011 had 2.238 times the odds of being *Staphylococcus aureus* compared to cases in other years. There were also statistically significant differences (*p* < 0.001) in the event numbers of diarrhea clusters among public places between 2011 and 2019 (Table 4). School was identified as OR = 1.983.

Regarding viral infections, norovirus infections were the most numerous (77.1%, 1810/2347) in diarrhea clusters, whereas rotavirus infections were the second most numerous (11.5%, 271/2347) in diarrhea clusters (Table 2). Regarding bacterial infections, *St**aphylococcus aureus* was the most numerous (35.5%, 104/293) in diarrhea clusters, whereas *Bacillus cereus* was the second most numerous (23.2%, 68/293) among diarrhea clusters (Table 3). Schools were the most common location (49.1%, 1406/2865) of diarrhea clusters, whereas the hospitality industry was the second most common location (22.9%, 656/2865) of diarrhea clusters (Table 4).

## 4. Discussion

Diarrhea is highly infectious, can be caused by a variety of bacteria, viruses, or parasites, and can easily lead to cluster infections. In Taiwan, diarrheal infectious diseases such as cholera, typhoid fever/paratyphoid fever, bacillary dysentery, amoebic dysentery, and enterohemorrhagic *E. coli* infection are notifiable diseases in accordance with the provisions of the Communicable Disease Control Act [25], Other pathogens causing diarrhea, such as norovirus, rotavirus, non-typhoidal *Salmonella*, non-toxigenic *Vibrio cholerae*, *Staphylococcus aureus*, *Cactus bacilli*, and *Vibrio parahaemolyticus* are not notifiable diseases, but they can still cause clustering events and are harmful to people.

Diarrhea is a symptom of gastrointestinal infection, which is mainly transmitted through the oral–fecal route and human-to-human contact, including failure to maintain good hygiene, close contact with a patient, contact or consumption of water contaminated by a patient’s excreta and vomit, food, or the inhalation of droplets from vomiting and infection, often causing large-scale clustering events [26]. During the investigation period of this study, the results indicate that there were statistically significant differences in the distribution of bacterial and viral pathogens among diarrhea cluster events in each year (*p* < 0.001). From 2011 to 2019, among all the pathogens causing diarrhea, the most numerous diarrhea clusters in each year were caused by norovirus. This study also similar to results found in the literature for other countries [27,28]. Rotavirus was the pathogen with the second highest number of diarrhea clusters. In Taiwan, the number of bacterial and viral clustering events were seen to increase over time. This study suggested that this might be due to the correlation with temperature or domestic water consumption. The annual average temperature was 21.06 °C in 2011, 22.10 °C in 2015, and 22.38 °C in 2019 in Taiwan [29]. As the temperature increases, it may accelerate the spoilage of food, and individuals may suffer from diarrhea. Moreover, daily domestic water consumption per person was 270 L in 2011, 273 L in 2015, and 284 L in 2019 in Taiwan [30]. The environment is hit by typhoons and extreme rainfall in Taiwan. As the quality of tap water deteriorates due to typhoons or excessive rainfall, it may cause diarrhea in individuals. These factors may not only increase the burden on public health and epidemic prevention personnel, but may also mean that physicians have difficulty making a differential diagnosis and waste medical resources. There may also be more symptom variability in diarrhea patients. The results of the current study may provide health authorities in Taiwan with control strategies to prevent the person-to-person transmission of norovirus, rotavirus and infectious bacteria.

In the U.S., Chai et al. reported that diarrhea cluster events caused by pathogenic bacteria between 1998 and 2013 were dominated by *Salmonella enterica* and *Closedium perfringen* [31]. However, in our study focusing on Taiwan from 2011 to 2019, the causes of the highest number of diarrhea cluster events were *Staphylococcus aureus* and *Bacillus cereus* (35.5% vs. 23.3%). This result may imply that national conditions, lifestyle, cleaning materials and eating habits are different in Taiwan and the United States, resulting in varied incidence rates of diarrhea clustering caused by diverse pathogens. Moreover, in this study, it appeared that there were statistically significant differences in the distribution of “bacterial pathogens” in the diarrhea cluster events of each year (*p* < 0.001). In addition, this study showed that there were more than 100 diarrhea clustering events caused by pathogenic bacteria in the nine-year period examined in this study (with most occurring in 2017–2019). Furthermore, the number of diarrhea clustering events caused by different pathogens varied according to year. From 2017 to 2019, pathogenic bacteria causing the greatest number of clusters were *Bacillus cereus* in 2017 and *Staphylococcus aureus* in 2018–2019. This inconsistent infection trend may indicate that the role of pathogens causing diarrhea clustering through interaction in the human environment, among the pathogenic bacteria in Taiwan, is more diverse and complicated. This is a serious issue for epidemic prevention, and Taiwan’s health department needs to further assess and/or develop public health intervention programs to prevent or effectively mitigate diarrheal clusters in the future.

The results of this study show that there are also statistically significant differences in the distribution of “institutions (or places)” in the diarrhea cluster events of each year. Schools accounted for 49.1% of all diarrheal cluster events according to the TNIDSS in Taiwan. According to the literature, diarrhea outbreaks in schools occur at a higher rate in China, Hong Kong and Japan [32,33,34]. This is different from Europe and the United States, where acute and long-term health care institutions (such as nursing homes and sanatoriums) have been identified as the most common environments for norovirus outbreaks [35,36,37]. The main environmental differences among foodborne diseases may reflect the greater quantity and larger scale of long-term care institutions in Europe and the United States in comparison with Taiwan. Each school in Taiwan has an average of 3550 students in each class, which may constitute a higher density of the population compared to other countries.

Close contact can increase the spread of human-to-human diseases among students, especially in children, who wash their hands less frequently. Student life on campus is group-oriented. Due to a common diet, water exposure and close contact, infectious disease clusters are at a higher risk of disseminating on campus, especially through droplets, feces and contact routes to spread pathogens. Acute diarrhea clustering often occurs on campuses. The clustering of diarrhea may be caused by pathogenic contamination from institutional food services, human-to-human contact, and water consumed by students.

According to statistics by Taiwan’s CDC, the frequency of viral gastroenteritis clustering on campus is higher than bacterial gastroenteritis and affects many students. The viral gastroenteritis pathogens (such as norovirus) causing food poisoning events have also caught the attention of government health agencies. The stool of someone with viral gastroenteritis is usually watery or a soft paste, with little blood or mucus, and the individual does not often have a high fever. However, bacterial infection is more prone to producing mucus, blood, and a high fever. When diarrhea clustering occurs in schools, it can be preliminarily distinguished by the symptoms, magnitude of the effects, and the season of onset. Therefore, it is necessary to enhance the awareness of teachers and students about acute diarrhea and understand preventive measures, so as to implement epidemic prevention and maintain the health of people on campus.

In recent years, due to an increasing awareness of health care, large-scale diarrhea cluster events have often concerned people. For example, during the Moon Festival in 2016, there was an outbreak in Toufen in Miaoli. One hundred and forty-three people had diarrhea caused by eating roast duck contaminated with *Salmonella* bought from the market [38]. In the face of large-scale cluster events or concern about cluster events, rapid diagnosis of the infection source is helpful for epidemic control. However, there were 2865 reported diarrhea clusters in Taiwan from 2011 to 2019, out of which the pathogens involved failed to be detected using routine tests in 380 cases (about 20%).

We suggest that non-routine pathogen detection (multiplex PCR detection and identification of pathogens by the culture of the special screening medium, etc.) may help to clarify other pathogens, increase the screening rate of pathogens, and improve testing efficacy. In addition, non-routine testing requires a higher cost, so that it should be preferentially applied to sudden, or large, cluster cases.

In order to effectively prevent norovirus diarrhea clustering and reduce the risk of disease (death) caused by pathogen transmission, we suggest that the specific approach of relevant institutions and populations should be as follows. (1) In general institutions (locations): people with diarrhea symptoms (especially students, young children, caregivers, and kitchen workers in the catering industry) should immediately stop processing food and wash hands thoroughly. Those who are in danger of infection should stay at home. If they need to go out, they should wear masks to avoid infecting others, and drink extra water and electrolytes and maintain good nutrition. Patients should return to work or school at least 48 h after the symptoms are relieved in order to reduce the risk of pathogen transmission on campus or in workplaces. Clothes, sheets and bedding contaminated by the vomit or excreta of suspected or confirmed cases should be changed immediately. People should wear gloves and masks and clean the environment with 100cc bleach (containing 5% sodium hypochlorite) in 1 L of water. The environment and surface of objects around the patient, such as the bedside and tabletop, can be wiped with 20cc bleach and 1 L of bleach. When a cluster of suspected diarrhea occurs in schools, institutions, and medical institutions, it should be immediately reported to the local health authority. Additionally, the patient should arrange medical treatment, or move to an independent or isolated space, cooperate with the health authority for specimen collection and epidemic investigation, and comply with other prevention measures. The school administrators should strengthen the management of food safety on campus and assign personnel to supervise the source of food materials supplied by the central kitchen. Environmental sanitation and conditioned food should meet health regulations. Enhanced health education and information, including good personal hygiene habits such as not eating uncleaned raw food, not drinking drinks made of raw ingredients, and washing hands frequently, should be provided. The catering industry should pay attention to the hygiene of food materials and food conditioning processes, including the hand hygiene management of catering employees, cooking of shellfish products, and separate treatment of raw and cooked food to avoid cross contamination. In densely populated institutions (e.g., health care institutions), staff should follow proper hand washing procedures before and after work and improve hand and personal hygiene. Hands should be washed before and after visiting residents. In medical institutions, infected patients should be transferred to a single ward, a general isolation ward or a designated area for centralized care, and standard protection and contact infection protection measures should be implemented. Hand washing procedures should be strictly implemented before and after coming into contact with patients. In terms of family living, people should wash their hands frequently after going to the toilet or before eating or preparing food; fresh ingredients should be selected for cooking, food hygiene and preservation measured should be adhered to, uncooked eggs, meat or raw shellfish should be avoided, and raw and cooked food should be treated separately. Before and after coming into contact with or treating patients, individuals should wash their hands thoroughly to reduce the incidence of infections in the family. Patients with norovirus should avoid visiting relatives and friends in hospitals or health care institutions to avoid spread in hospitals or institutions.

There are several limitations to this study. (1) The statistical data about infectious diseases published by Taiwan’s CDC on the online platform only provide basic epidemiological data about diarrhea clustering events, without clinical data. Therefore, this study cannot compare differences or trends in clinical symptoms. (2) There is no information about norovirus or rotavirus strains or the genotype of *Staphylococcus aureus* on the platform, so this study does not show (a) the strain of Taiwan norovirus or rotavirus or the genotype of *Staphylococcus aureus* that is prevalent, or (b) the genetic relationship when comparing the virus strains in Taiwan with other countries or comparing the genotype of *Staphylococcus aureus*. The results of this study suggest that the government should collect or publish more detailed information on the pathological causes and pathogenic genes of foodborne diseases, and allow researchers to make in-depth use of more complete data and analysis. (3) Based on the data from Taiwan’s CDC, there is also no information about the data available on pathogens and setting and year, which cannot be presented in this study. We are not sure whether the surveillance has changed, but we will continue to follow up the government’s health policy about preventing diarrhea clusters. This study has one advantage, namely, the existing online platform of Taiwan’s health department has stored data on diarrhea clusters over many years, which enables public health (clinical) researchers or institutions to analyze data and enhance the research energy of local infectious disease epidemiology.

To our knowledge, this is the first study to explore the epidemiological characteristics of diarrhea clusters over the period of 2011 to 2019 using the government’s open dataset in Taiwan. This study confirmed that high risk places include those where food or drink are consumed (schools, populous institutions and hospitality industry venues) and etiological substances (norovirus, norovirus and rotavirus coinfection, rotavirus, *Staphylococcus aureus*, *Vibrio parahaemolyticus*, *Salmonella*, *Bacillus cereus*) may lead to diarrhea clusters. In addition, the number of diarrhea clusters showed an increasing trend year by year in schools and for pathogenic viruses (norovirus) and pathogenic bacteria (*Bacillus cereus*), and a decreasing trend for *Salmonella* infections.

## 5. Conclusions

These results may be worthy of attention from the government’s health department for decision-making or as a reference for clinical or epidemiological expert research. With the current findings in mind, we recommend that government departments continue to develop new laboratory techniques and diagnostic standards to increase knowledge about the epidemiological characteristics of diseases and provide data to track diarrhea clusters and discuss the epidemic patterns, trends and risk factors in the future.

## Figures and Tables

**Table 1 children-08-00807-t001:** Descriptive statistics of diarrhea clusters in Taiwan, 2011–2019.

Parameter	Year								
	2011 N ^a^ = 114 (%)	2012 N = 145 (%)	2013 N = 106 (%)	2014 N = 106 (%)	2015 N = 357 (%)	2016 N = 452 (%)	2017 N = 575 (%)	2018 N = 424 (%)	2019 N = 586 (%)
^b^ Positive	95 (83.3)	112 (77.2)	68 (64.2)	63 (59.4)	274 (76.8)	357 (79.0)	494 (85.9)	308 (72.6)	425 (72.5)
^c^ Negative	18 (15.8)	32 (22.1)	37 (34.9)	43 (40.6)	81 (22.7)	94 (20.8)	75 (13.0)	110 (25.9)	160 (27.3)
No specimen	1 (0.9)	1 (0.7)	1 (0.9)	-	2 (0.5)	1 (0.2)	6 (1.0)	6 (1.4)	1 (0.2)

^a^: event numbers; ^b^: Routine testing, pathogens were detected; ^c^: Routine testing, no pathogens were detected, -: not application.

**Table 2 children-08-00807-t002:** Characteristics of pathogens causing diarrhea clusters in Taiwan, 2011–2019.

Pathogen		Year									*p*
	Overal N = 2196 (%)	2011 N ^a^ = 95 (%)	2012 N = 112 (%)	2013 N = 68 (%)	2014 N = 63 (%)	2015 N = 274 (%)	2016 N = 357 (%)	2017 N = 494 (%)	2018 N = 308 (%)	2019 N = 425 (%)	
Norovirus ^b^	1659 (75.5)	70 (73.7)	104 (92.9)	53 (77.9)	42 (66.7)	197 (71.9)	298 (83.5)	345 (69.8)	202 (65.6)	348 (81.9)	<0.001
Norovirus + Rotavirus	151 (6.9)	11 (11.6)	2 (1.8)	3 (4.4)	11 (17.5)	21 (7.7)	17 (4.8)	63 (12.8)	22 (7.1)	1 (0.2)	
Rotavirus	120 (5.5)	6 (6.3)	3 (2.7)	11 (16.2)	8 (12.7)	17 (6.2)	12 (3.4)	41 (8.3)	22 (7.1)	-	
Adenovirus	1 (<0.1)	-	-	-	-	-	-	1 (0.2)	-	-	
Bacterial pathogen	265 (12.1)	8 (8.4)	3 (2.7)	1 (1.5)	2 (3.2)	39 (14.2)	30 (8.4)	44 (8.9)	62 (20.1)	76 (17.9)	

^a^: event numbers; ^b^: Norovirus single infection (odds ratio, OR = 4.423, *p* < 0.001) in 2012 compared with other years, -: not application.

**Table 3 children-08-00807-t003:** Bacteria and other pathogens causing diarrhea clusters in Taiwan, 2011–2019.

Pathogen		Year									*p*
	Overal N = 265 (%)	2011 N = 8 ^a^ (%)	2012 N = 3 (%)	2013 N = 1 (%)	2014 N = 2 (%)	2015 N = 39 (%)	2016 N = 30 (%)	2017 N = 44 (%)	2018 N = 62 (%)	2019 N = 76 (%)	
*Staphylococcus aureus* ^b^	84 (31.7)	4 (50)	-	1 (100)	1 (50)	11 (28.2)	5 (16.7)	10 (22.7)	26 (41.9)	26 (34.2)	<0.001
*Vibrio parahaemolyticus*	45 (17.0)	2 (25)	-	-	-	11 (28.2)	6 (20)	13 (29.5)	5 (8.1)	8 (10.5)	
*Salmonella*	58 (21.9)	1 (12.5)	1 (33.3)	-	1 (50)	10 (25.6)	12 (40)	4 (9.1)	8 (12.9)	21 (27.6)	
*Bacillus cereus*	47 (17.7)	-	-	-	-	3 (7.7)	7 (23.3)	16 (36.4)	13 (21.0)	8 (10.5)	
*Shigella dysenteriae*	3 (1.1)	1 (12.5)	2 (66.7)	-	-	-	-	-	-	-	
*Vibrio cholerae* (Non-O1,Non-O139)	2 (0.7)	-	-	-	-	1 (2.6)	-	-	1 (1.6)	-	
*Staphylococcus aureus* + *Vibrio cholerae* (Non-O1,Non-O139)	2 (0.7)	-	-	-	-	1 (2.6)	-	-	1 (1.6)	-	
*Vibrio parahaemolyticus* + *Bacillus cereus*	1 (0.4)	-	-	-	-	1 (2.6)	-	-	-	-	
*Staphylococcus aureus* + *Salmonella*	1 (0.4)	-	-	-	-	1 (2.6)	-	-	-	-	
*Staphylococcus aureus* + *Bacillus cereus*	13 (4.9)	-	-	-	-	-	-	1 (2.3)	3 (4.8)	9 (11.8)	
*Staphylococcus aureus* + *Vibrio parahaemolyticus*	2 (0.7)	-	-	-	-	-	-	-	-	2 (2.6)	
*Bacillus cereus* + *Salmonella*	4 (1.5)	-	-	-	-	-	-	-	2 (3.2)	2 (2.6)	
*Bacillus cereus + Salmonella + Staphylococcus aureus*	2 (0.7)	-	-	-	-	-	-	-	2 (3.2)	-	
*Bacillus cereus + Vibrio cholera Non-O1, Non-O139*	1 (0.4)	-	-	-	-	-	-	-	1 (1.6)	-	

^a^: event numbers. ^b^: *Staphylococcus aureus* single infection (odds ratio, OR = 2.238, *p* = 0.252) in 2011 compared with other years, -: not application.

**Table 4 children-08-00807-t004:** Institutions where diarrhea clusters took place in Taiwan, 2011–2019.

Institution		Year									*p*
	Overal N = 2865 (%)	2011 N ^a^ = 114 (%)	2012 N = 145 (%)	2013 N = 106 (%)	2014 N = 106 (%)	2015 N = 357 (%)	2016 N = 452 (%)	2017 N = 575 (%)	2018 N = 424 (%)	2019 N = 586 (%)	
School	1406 (49.1)	70 (61.4)	74 (51.0)	69 (65.1)	68 (64.2)	159 (44.5)	182 (40.3)	308 (53.6)	193 (45.5)	283 (48.3)	<0.001
Hospitality Industry ^b^	656 (22.9)	-	-	-	-	100 (28.0)	155 (34.3)	124 (21.6)	119 (28.1)	158 (27.0)	
Populous Institutions ^c^	295 (10.3)	19 (16.7)	41 (28.3)	20 (18.9)	24 (22.6)	30 (8.4)	38 (8.4)	36 (6.3)	41 (9.7)	46 (7.8)	
Hospital	120 (4.2)	12 (10.5)	20 (13.8)	7 (6.6)	5 (4.7)	25 (7.0)	8 (1.8)	11 (1.9)	13 (3.1)	19 (3.2)	
Military Camp	45 (1.6)	2 (1.8)	2 (1.4)	4 (3.8)	1 (0.9)	5 (1.4)	6 (1.3)	16 (2.8)	3 (0.7)	6 (1.0)	
Other ^d^	343 (12.0)	11 (9.6)	8 (5.5)	6 (5.7)	8 (7.5)	38 (10.6)	63 (13.9)	80 (13.9)	55 (13.0)	74 (12.6)	

^a^: event numbers. ^b^: included accommodation and food service activities, restaurant, hotel, etc. ^c^: included general nursing homes, mental nursing homes, elderly welfare institutions, long-term care service institutions, honorary national homes, disability welfare institutions, mental rehabilitation institutions, children and youth placement and correctional institutions, correctional institutions, etc. ^d^ Others: includes places of business, tour groups, family, dormitories and camps, -: not application. School-related diarrhea clusters (odds ratio, OR = 1.983, *p* = 0.001) in 2013 compared with other years.

## Data Availability

Taiwan Centers for Disease Control. Taiwan National Infectious Disease Statistics System. Available online: https://nidss.cdc.gov.tw/ch/ (Accessed on: 1 July 2021).

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
