# Peer review of "An Increased Risk of School-Aged Children with Viral Infection among Diarrhea Clusters in Taiwan during 2011–2019"

_children, 2021, doi:10.3390/children8090807_

Round 1
Reviewer 1 Report
Comments to Manuscript ID: children-1322485
Overall: define wha you mean by cluster and why it is not the same as an outbreak.
Line 37: in the abstract you state that noro- and rotavirus are the main causes, here you sate it is enteroviruses. Please be consistent.
Line 39: change ‘may be’ to ‘is’.
Line 40: change ‘aa’ to ‘a’.
Line 55-56: update reference to PMID: 31483239 (ten genogroups)
Line 66 and 216: Change ‘trans-mission’ to ‘transmission’
Line 111: Change ‘do-main’ to ‘domain’.
Line 114: Change ‘ac-companied’ to ‘accompanied’
Line 114: Viral diarrhea is normally not linked to bloody diarrhea
Line 139: Change ‘pro-vided’ to ‘provided’
Line 173: Change ‘Stsphylococcus aureus’ to ‘Staphylococcus aureus’ and write in cursive
Line 198: Change ‘Ac-cording’ to ‘According’
Line 208 and 215: Change ‘Tai-wan’ to ‘Taiwan’
Line 211: Change ‘in-crease’ to ‘increase’
Line 258: Change ‘How-ever’ to ‘However’
Please condense the discussion, it is too long an repetitive
Line 398: Change ‘dis-cuss’ to ‘discuss’
Author Response
Dear the reviewers: August 5, 2021
We resubmitted the manuscript entitled “An increased risk of school-aged children with viral infection among diarrhea clusters in Taiwan during 2011-2019” to the Journal after amendments made based on reviewers comments. We have carefully revised our manuscript according to reviewer’s critiques and suggestions. We marked amendments in yellow font in the manuscript for clarity purpose. Our specific responses to reviewer’s comments are as follows.
Reviewer 1
- Overall: define wha you mean by cluster and why it is not the same as an outbreak.
Response: Thank you very much for your comments. Based on website data of CDC, USA (https://www.cdc.gov/csels/dsepd/ss1978/lesson1/section11.html), it indicated that outbreak carries the same definition of epidemic (Epidemic refers to an increase, often sudden, in the number of cases of a disease above what is normally expected in that population in that area), but is often used for a more limited geographic area. Cluster refers to an aggregation of cases grouped in place and time that are suspected to be greater than the number expected, even though the expected number may not be known. Conceptual Framework of This study belongs to the latter. The authors have added the sentence in the revised manuscript. Please see line 116-118.
- Line 37: in the abstract you state that noro- and rotavirus are the main causes, here you sate it is enteroviruses. Please be consistent.
Response: We have changed to “norovirus” and “rotavirus”. Please see line 38.
- Line 39: change ‘may be’ to ‘is’.
Response: We have replaced “may be” with “is”. Please see line 39.
Line 40: change ‘aa’ to ‘a’.
Response: We have replaced “aa” with “a”. Please see line 40.
Line 55-56: update reference to PMID: 31483239 (ten genogroups)
Response: We have update reference to PMID: 31483239 (ten genogroups). Please see line 54 and reference 4.
Line 66 and 216: Change ‘trans-mission’ to ‘transmission’
Response: We have replaced “trans-mission” with “transmission”. Please see line 64 and line 208.
Line 111: Change ‘do-main’ to ‘domain’.
Response: We have replaced “do-main” with “domain”. Please see line 109.
Line 114: Change ‘ac-companied’ to ‘accompanied’
Response: We have replaced “ac-companied” with “accompanied”. Please see line 112.
Line 114: Viral diarrhea is normally not linked to bloody diarrhea
Response: We have deleted “or mucus or blood-tinged in stool”. Please see line 113.
Line 139: Change ‘pro-vided’ to ‘provided’
Response: We have replaced “pro-vided” with “provided”. Please see line 143.
Line 173: Change ‘Stsphylococcus aureus’ to ‘Staphylococcus aureus’ and write in cursive
Response: We have replaced “Stsphylococcus aureus” with “Staphylococcus aureus”. Please see line 177.
Line 198: Change ‘Ac-cording’ to ‘According’
Response: Because the other reviewer comments, we have deleted the sentence as “According…”. Please see line 194.
Line 208 and 215: Change ‘Tai-wan’ to ‘Taiwan’
Response: Because the other reviewer comments, we have deleted the sentence as “…Taiwan…” in line 208. Please see line 198 of revised manuscript. However line 214 have changed Taiwan. Please see line 207.
Line 211: Change ‘in-crease’ to ‘increase’
Response: We have replaced “in-crease” with “increase”. Please see line 203.
Line 258: Change ‘How-ever’ to ‘However’
Response: We have replaced “How-ever” with “However”. Please see line 250.
Please condense the discussion, it is too long an repetitive
Response: We have deleted two paragraph as “It should be noted that…the ravages of enteric virus” and “Another issue of global concern…health policies and health resource allocation”. Please see line 309-310.
Line 398: Change ‘dis-cuss’ to ‘discuss’
Response: We have replaced “dis-cuss” with “discuss”. Please see line 338.
Hopefully, our revised manuscript could fulfill your scientific requirements for publication.
Sincerely yours,
Chia-Peng Yu, Ph.D. (the corresponding author)
School of Public Health,
National Defense Medical Center
No.161 Sec. 6, Minquan E. Rd., Neihu Dist., Taipei 114, Taiwan, Republic of China,
Tel: +886-2-87923311 ext. 16791, Fax: +886-2-87924379,
e-mail: yu6641@gmail.com

Reviewer 2 Report
Summary
Lin et al. use publicly available data from Taiwan CDC to describe the etiologies and locations of diarrhea cluster events reported from 2011 through 2019. Such information could be useful to inform public health action and research. However, I have some major concerns with this analysis. The first concern is whether there was a change in surveillance practices during the analysis period—beginning in 2015, there is a large jump in the number of events detected. This suggests that surveillance might have improved and become more sensitive, which would bias comparisons. I would request that the authors examine this possibility, perhaps by consulting experts at Taiwan CDC to ensure a complete understanding of the data. It might be possible/appropriate to restrict the analysis to 2015 – 2019. A second concern is about the analytic methods. I suggest engaging a statistician to confirm and explain the methods, since some of the methods seem inappropriate (e.g., chi square when Fisher is needed, logistic regression model with unclear outcome). Generally, the authors are advised to more completely and clearly describe all the methods, and also to focus on what is most relevant to the analysis and implications. Specific comments are below.
Introduction
- In lines 37-38, the authors mention that “infectious enterovirus is the main cause of the disease,” but I think they are referring more generally to enteric viruses, rather than enterovirus, since enterovirus ranks below other viral causes of diarrhea
- Suggest to update lines 53 – 54 with Chhabra 2019
- I am not sure it’s fully correct to say that the lack of long-lasting immunity to norovirus is primarily a result of the mutation of the virus. There is still a lot that is unknown about immunity to norovirus, with estimates for duration of homotypic immunity ranging from 1 to ~5 years.
- See Chhabra 2021 in CID and Simmons 2013 in EID
- Lines 84 – 88: I suggest updating these references and including more global data
- Global Burden of Disease has estimates for 2016 mortality
- A recent meta analysis by Burnett et al. shows impact estimates for rotavirus vaccine and includes summaries of rotavirus positivity for hospitalized children with AGE
- Alternatively, you could use data from Taiwan
- The reason for including multiple references from Australia is unclear to me
- Line 99, replace “occurred” with “was identified”
Methods
- Please define a “diarrhea clustering event” – is this based on number of persons with diarrhea in a similar time / place? Same etiology?
- Please clarify whether the unit of analysis (event) is a person with diarrhea or an outbreak of diarrhea (i.e., multiple persons sharing a time/ place)
- Are diarrhea clustering events reportable in Taiwan? If not, how can we know that the data included are representative of all diarrhea clustering events ?
- For some comparisons, Fishers’ Exact Test will be necessary due to the small numbers
- I don’t know that Odds Ratio is an appropriate method given the data structure and research questions. You might instead consider Poisson or Negative Binomial models since your outcome is a count (diarrhea clustering event)
- Either way, please define your models in the Methods (predictors, outcomes, model structure) so that the reader can better interpret the results
- The last 2 sentences are redundant
Results
- Figures 1 and 2 are a bit redundant with Tables 2 and 3, and also a bit confusingly presented. I would suggest that the information just be added to Tables 2 and 3 as an “overall” column
- Similarly, Figure 3’s information can be included in Table 4, and then Figure 3 can be deleted
- Table 3 is missing some %s
Discussion
- The significance of the week March 3 – 9, 2019, discussed in lines 198 – 203, is not clear to me. Why mention this?
- In lines 207 – 208, why focus on 2018 – 2019? These are included in the study period. Suggest removing this part of the sentence and instead focusing on the comparison of Taiwan to other countries
- Please support the statement of line 210 – 211 that “In Taiwan, the number of bacterial and viral clustering events may increase by about 5-6 times over years”
- I can see that there is an upward trend in number of clustering events in your data, but it is not yet clear to me that this is a true indicator of increased clustering events. You will need to rule out the possibility that surveillance of clustering events has changed over time (for instance, in 2015, when the number of clustering events suddenly jumped)
- I would advise caution in interpreting changes in prevalence of bacterial pathogens (or events by setting) by year, given the small numbers. It also seems that there was a big jump in 2015 that might be related to changes in surveillance
- Generally, the Discussion can be streamlined to focus more on the results of the analysis, how to interpret and compare them to other studies, and the public health implications. Some of the information provided in the Discussion is repetitive with what was covered in the Introduction, and could be taken out
Author Response
Dear the reviewers: August 5, 2021
We resubmitted the manuscript entitled “An increased risk of school-aged children with viral infection among diarrhea clusters in Taiwan during 2011-2019” to the Journal after amendments made based on reviewers comments. We have carefully revised our manuscript according to reviewer’s critiques and suggestions. We marked amendments in yellow font in the manuscript for clarity purpose. Our specific responses to reviewer’s comments are as follows.
Reviewer 2
- Lin et al. use publicly available data from Taiwan CDC to describe the etiologies and locations of diarrhea cluster events reported from 2011 through 2019. Such information could be useful to inform public health action and research. However, I have some major concerns with this analysis. The first concern is whether there was a change in surveillance practices during the analysis period—beginning in 2015, there is a large jump in the number of events detected. This suggests that surveillance might have improved and become more sensitive, which would bias comparisons. I would request that the authors examine this possibility, perhaps by consulting experts at Taiwan CDC to ensure a complete understanding of the data. It might be possible/appropriate to restrict the analysis to 2015 – 2019. A second concern is about the analytic methods. I suggest engaging a statistician to confirm and explain the methods, since some of the methods seem inappropriate (e.g., chi square when Fisher is needed, logistic regression model with unclear outcome). Generally, the authors are advised to more completely and clearly describe all the methods, and also to focus on what is most relevant to the analysis and implications. Specific comments are below.
Response:
- The authors agree the reviewer opinions that there was a change in surveillance practices during the analysis period—beginning in 2015, there is a large jump in the number of events detected. This suggests that surveillance might have improved and become more sensitive. In the next 5 or 5-10 years of diarrhea cluster research, the authors will definitely adopt the opinions of the reviewer. Thanks the reviewer comment.
- We would agree the opinions of the reviewer. We have engaged a statistician to confirm and explain the methods, revised some of the methods as used Fisher’s exact probability test. Thanks the reviewer comment. Please see line 134-138.
Introduction
- In lines 37-38, the authors mention that “infectious enterovirus is the main cause of the disease,” but I think they are referring more generally to enteric viruses, rather than enterovirus, since enterovirus ranks below other viral causes of diarrhea
Response: We have changed to “norovirus” and “rotavirus”. Please see line 38
- Suggest to update lines 53 – 54 with Chhabra 2019
Response: We have update reference to PMID: 31483239 (ten genogroups). Please see line 54 and reference 4.
- I am not sure it’s fully correct to say that the lack of long-lasting immunity to norovirus is primarily a result of the mutation of the virus. There is still a lot that is unknown about immunity to norovirus,
See Chhabra 2021 in CID and Simmons 2013 in EID
Response:
The authors have agreed the reviewer comment. We cancelled the sentence as” Therefore, it is difficult for the human immune system to produce long-term protection against norovirus”. Please see line 59.
- Lines 84 – 88: I suggest updating these references and including more global data
Global Burden of Disease has estimates for 2016 mortality
A recent meta analysis by Burnett et al. shows impact estimates for rotavirus vaccine and includes summaries of rotavirus positivity for hospitalized children with AGE
Alternatively, you could use data from Taiwan
The reason for including multiple references from Australia is unclear to me
Response:
We updating these references and the line 84-88 of original manuscript have deleted. We have revised the sentence as “The report of the Global Burden of Disease and several extended analyses on rotavirus and results of rotavirus vaccination found that rotavirus infection caused 128 500 deaths and 258 173 300 episodes of diarrhea among children younger than 5 years in 2016 [18]. A recent meta analysis by Burnett et al. shows impact estimates for rotavirus vaccine and includes summaries of rotavirus positivity for hospitalized children with AGE [19]”. Please see line 82-87
- Line 99, replace “occurred” with “was identified”
Response:
We have replaced “occoured” with “was identified”. Please see line 97.
Methods
- Please define a “diarrhea clustering event” – is this based on number of persons with diarrhea in a similar time / place? Same etiology?
Response:
The authors have agreed the reviewer comment. The authors have revised the sentence as line 115-117 in revised manuscript.
- Please clarify whether the unit of analysis (event) is a person with diarrhea or an outbreak of diarrhea (i.e., multiple persons sharing a time/ place)
Response:
The authors have agreed the reviewer comment. The diarrhea clustering event is based on number of multiple persons with diarrhea in a similar time / place and patients with same pathogens infection. The authors have added the sentence. Please see line 115-117.
- Are diarrhea clustering events reportable in Taiwan? If not, how can we know that the data included are representative of all diarrhea clustering events?
Response:
The diarrhea clustering events should report in Taiwan. Based on Taiwan’s CDC policy, patients of suspected food poisoning events that report to product management distribution system (PMDS) and obtained the event ID, are able to submit specimens through reporting to the Symptom Surveillance System. The authors have added the sentence. Please see line 117-120.
- For some comparisons, Fishers’ Exact Test will be necessary due to the small numbers
Response:
We would agree the opinions of the reviewer. We have engaged a statistician to confirm and explain the methods, revised some of the methods as used Fisher’s exact probability test. Please see line 134-135
- I don’t know that Odds Ratio is an appropriate method given the data structure and research questions. You might instead consider Poisson or Negative Binomial models since your outcome is a count (diarrhea clustering event)
Response:
Thanks the reviewer comment and suggestion. We had consulted a statistician to confirm and explain the methods. We agreed with that the Poisson or Negative Binomial models is an appropriate method for a count data, but it is not the focus of this research. We concerned the relationship of binary outcome (e.g., Norovirus vs. non- Norovirus, Staphylococcus aureus vs. non- Staphylococcus aureus, school vs. non-school) with the other risk factor. We think the Odds Ratio by logistic regression is also the appropriate methods given the data structure and research questions. Please see line 135-138.
- Either way, please define your models in the Methods (predictors, outcomes, model structure) so that the reader can better interpret the results
Response:
Thanks the reviewer comment and suggestion. We had clearly defined our models in the Methods to help the reader interpret the results easily. Please see line 135-138. The models in the Methods of this study is the same as our previous publication [1, 2].
Reference 1.
Yu-Ching Chou, Chi-Jeng Hsieh, Chun-An Cheng, Ding-Chung Wu, Wen-Chih Wu, Fu-Huang Lin, Chia-Peng Yu. Epidemiologic Characteristics of Imported and Domestic Chikungunya Cases in Taiwan: A 13-Year Retrospective Study. Int J Environ Res Public Health. 2020 May; 17(10): 3615.
Reference 2.
Chi-Jeng Hsieh, Chuan-Wang Li, Chun-An Cheng, Ding-Chung Wu, Wen-Chih Wu, Fu-Huang Lin, Yu-Ching Chou, Chia-Peng Yu. Epidemiologic Characteristics of Domestic Patients with Hemorrhagic Fever with Renal Syndrome in Taiwan: A 19-Year Retrospective Study. Int J Environ Res Public Health. 2020 Aug; 17(15): 5291.
- The last 2 sentences are redundant
Response:
The authors have deleted the last 2 sentences. Please see line 139.
Results
- Figures 1 and 2 are a bit redundant with Tables 2 and 3, and also a bit confusingly presented. I would suggest that the information just be added to Tables 2 and 3 as an “overall” column
Response:
Thanks the reviewer comment. The authors have agreed the reviewer comment. The information of Figures 1 and 2 is added to Tables 2 and 3 as an “overall” column. Then, Figure 1 and 2 have deleted. Please see page 4-5 (Table 2-3 in revised manuscript).
- Similarly, Figure 3’s information can be included in Table 4, and then Figure 3 can be deleted
Response:
Thanks the reviewer comment. The authors have agreed the reviewer comment. The information of Figures 3 is added to Tables 4 as an “overall” column. Figure 3 have deleted. Please see page 6 (Table 4 in revised manuscript).
- Table 3 is missing some %s
Response:
The authors have added the data % in table 2, table 3 and table 4.
Discussion
- The significance of the week March 3 – 9, 2019, discussed in lines 198 – 203, is not clear to me. Why mention this?
Response:
Thanks the reviewer comment. The authors have deleted the sentence as lines 198-203. Please see line 194 in revised manuscript.
- In lines 207 – 208, why focus on 2018 – 2019? These are included in the study period. Suggest removing this part of the sentence and instead focusing on the comparison of Taiwan to other countries
Response:
Thanks the reviewer comment. The authors have revised the sentence as “This study is also similar to results found in the literature for other countries”. Please see line 199-200.
- Please support the statement of line 210 – 211 that “In Taiwan, the number of bacterial and viral clustering events may increase by about 5-6 times over years” I can see that there is an upward trend in number of clustering events in your data, but it is not yet clear to me that this is a true indicator of increased clustering events. You will need to rule out the possibility that surveillance of clustering events has changed over time (for instance, in 2015, when the number of clustering events suddenly jumped). I would advise caution in interpreting changes in prevalence of bacterial pathogens (or events by setting) by year, given the small numbers. It also seems that there was a big jump in 2015 that might be related to changes in surveillance
Response:
Thanks the reviewer comment. There is an upward trend in number of clustering events in this study. I have revised as “In Taiwan, the number of bacterial and viral clustering events may increase over years. It also seems that there was a big jump in 2015 that might be related to changes in surveillance.” Please see line 201-203.
- Generally, the Discussion can be streamlined to focus more on the results of the analysis, how to interpret and compare them to other studies, and the public health implications. Some of the information provided in the Discussion is repetitive with what was covered in the Introduction, and could be taken out
Thanks the reviewer comment. We have deleted the last two paragraph as “It should be noted that…the ravages of enteric virus” and “Another issue of global concern…health policies and health resource allocation”. Please see line 309.
Hopefully, our revised manuscript could fulfill your scientific requirements for publication.
Sincerely yours,
Chia-Peng Yu, Ph.D. (the corresponding author)
School of Public Health,
National Defense Medical Center
No.161 Sec. 6, Minquan E. Rd., Neihu Dist., Taipei 114, Taiwan, Republic of China,
Tel: +886-2-87923311 ext. 16791, Fax: +886-2-87924379,
e-mail: yu6641@gmail.com

Round 2
Reviewer 2 Report
I thank the authors for their revisions but I still have some concerns.
- Lines 85 - 86: Please describe the findings in Burnett et al rather than referring the reader to the paper
- Lines 94-95: Again, I don't know why Australia is discussed
- Please check references throughout--numbering has gotten out of order
- The unit of analysis is still unclear to me. Were the data organized by outbreak/cluster (i.e., containing multiple diarrhea cases, all tied to a single event) or by case (i.e., one person per observation)? If organized by cluster, then it would also be informative to analyze the number of cases per cluster. If organized by case, then you will have to account for clustering by case in your analytic methods.
- Line 160: "Norovirus single infection was confirmed as odds ratio..." -- This should be deleted. I think what you mean is that cases in 2012 had 4.23 times the odds of being norovirus as cases in other years.
- Similar comment for line 164 about S. aureus
- Were there data available on pathogen and setting and year? It would be informative to look at pathogens by setting
- Line 202: "a big jump" is very colloquial language. I suggest modifying this a bit and then also supporting this statement. Was there a change in surveillance? If you are not sure, you should discuss this in the limitations. Also, instead of "may increase over the years," which implies that there is an ongoing trend, you should instead speak to what you have observed--i.e., that the number increased over the study period, especially in 2015
- Line 275 - also please mention washing hands thoroughly
- More work needs to be done to clarify the methods
Author Response
Dear the reviewer, August 10, 2021
We resubmitted the manuscript entitled “An increased risk of school-aged children with viral infection among diarrhea clusters in Taiwan during 2011-2019” to the Journal after amendments made based on reviewers’ comments. We have carefully revised our manuscript according to reviewers’ critiques and suggestions. We marked amendments in yellow font in the manuscript for clarity purpose. Our specific responses to reviewers’ comments are as follows.
Reviewer 2
- Lines 85 - 86: Please describe the findings in Burnett et al rather than referring the reader to the paper
Response: The authors have agreed with the reviewer’s comment and revised the sentences in line 85-87 in the revised manuscript.
- Lines 94-95: Again, I don't know why Australia is discussed
Response: We have deleted the sentence as “the Australian…[19]”. Please see line 95.
- Please check references throughout--numbering has gotten out of order
Response: Thank you for the review. We have checked references throughout and make sure they are in order.
- The unit of analysis is still unclear to me. Were the data organized by outbreak/cluster (i.e., containing multiple diarrhea cases, all tied to a single event) or by case (i.e., one person per observation)? If organized by cluster, then it would also be informative to analyze the number of cases per cluster. If organized by case, then you will have to account for clustering by case in your analytic methods.
Response: Thank you for the comment. We have added sentence as “The data were organized by cluster which containing multiple diarrhea cases, all tied to a single event.” Please see line 116-117. Based on Taiwan’s CDC data, there are no information about the number of cases per diarrhea cluster, which are not able to be shown in this study. Please refer to limitations (line 327-329) in the revised manuscript.
- Line 160: "Norovirus single infection was confirmed as odds ratio..." -- This should be deleted. I think what you mean is that cases in 2012 had 4.23 times the odds of being norovirus as cases in other years.
Response: Thank you for your understanding. We indeed mean as your comment, and have revised the sentence as “Cases in 2012 had 4.423 times the odds of being norovirus as cases in other years.” Please see line 160-161.
- Similar comment for line 164 about S. aureus
Response: We have revised the sentence as “Cases in 2011 had 2.238 times the odds of being Staphylococcus aureus as cases in other years.” Please see line 164.
- Were there data available on pathogen and setting and year? It would be informative to look at pathogens by setting
Response: Based on Taiwan’s CDC data, there is no information about the data available on pathogen and setting and year, which are not able to be presented in this study. Please refer to limitations (line 327-329) in the revised manuscript.
- Line 202: "a big jump" is very colloquial language. I suggest modifying this a bit and then also supporting this statement. Was there a change in surveillance? If you are not sure, you should discuss this in the limitations. Also, instead of "may increase over the years," which implies that there is an ongoing trend, you should instead speak to what you have observed--i.e., that the number increased over the study period, especially in 2015.
Response: Thank you for the comment. We are not sure whether the surveillance have changed, but we will continue to follow up the government's health policy about preventing diarrhea clusters. Please refer to limitations on line 329-330 in the revised manuscript. We have added “This study suggested that it might be due to the correlation with temperature or domestic water consumption. The annual average temperature is 21.06℃ in 2011, 22.10℃ in 2015, and 22.38℃ in 2019 in Taiwan [29]. As the temperature increases, it may accelerate the spoilage of food, and individuals may suffer from diarrhea. Moreover, daily domestic water consumption per person is 270 liter in 2011, 273 liter in 2015, and 284 liter in 2019 in Taiwan [30]. The environment is hit by typhoons and extreme rainfall in Taiwan. As the quality of tap water deteriorates due to typhoon or excessive rainfall, it may cause diarrhea in individuals.” Please see line 201-209.
- Line 275 - also please mention washing hands thoroughly
Response: We have added “… and wash hands thoroughly”. Please see line 280.
- More work needs to be done to clarify the methods
Response: Thank you for the reviewer’s comment. We have clarified the methods of revised manuscript. We will pay further attention to the description of method in future research.
Hopefully, our revised manuscript could fulfill your scientific requirements for publication.
Sincerely yours,
Chia-Peng Yu, Ph.D. (the corresponding author)
School of Public Health,
National Defense Medical Center
No.161 Sec. 6, Minquan E. Rd., Neihu Dist., Taipei 114, Taiwan, Republic of China,
Tel: +886-2-87923311 ext. 16791, Fax: +886-2-87924379,
e-mail: yu6641@gmail.com
